# MSA GENERATION WITH SEQS2SEQS PRETRAINING: ADVANCING PROTEIN STRUCTURE PREDICTIONS

## ABSTRACT

Deep learning, epitomized by models like AlphaFold2 (Jumper et al., 2021), has achieved unparalleled accuracy in protein structure prediction. However, the depth of multiple sequence alignment (MSA) remains a bottleneck, especially for proteins lacking extensive homologous families. Addressing this, we present MSA-Generator, a self-supervised generative protein language model, pre-trained on a sequence**s**-to-sequence**s** task with an automatically constructed dataset. Equipped with protein-specific attention mechanisms, MSA-Generator harnesses large-scale protein databases to generate virtual, informative MSAs, enriching subpar MSAs and amplifying prediction accuracy. Our experiments with CASP14 and CASP15 benchmarks showcase marked LDDT improvements, especially for challenging sequences, enhancing both AlphaFold2 and RoseTTAFold's performance.

## 1 INTRODUCTION

The challenge of protein structure prediction (PSP), a pivotal issue in structural biology, has has experienced transformative progress due to the deep learning revolution. Among the advancements, AlphaFold2 (AF2) (Jumper et al., 2021) shines particularly bright, credited chiefly to its adept use of multiple sequence alignments (MSAs). MSAs, derived from querying a protein sequence against vast databases using search algorithms, represent aggregations of homologous sequences that capture evolutionary information, acting as the foundation of many PSP models. However, not all protein sequences possess a rich set of homologous counterparts. This scarcity often means that even advanced search algorithms struggle to construct high-quality MSAs, leading to a compromised efficacy for MSA-reliant models like AF2 (Jumper et al., 2021; Wang et al., 2022) as illustrated in fig. 1.

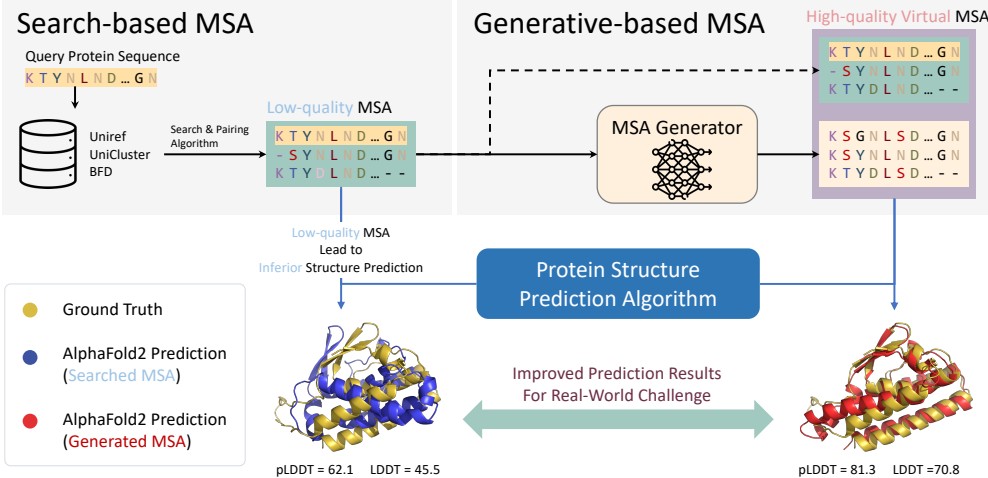

Figure 1: (**Top Left**) Certain protein sequences lack rich homologs, leading to poor MSA quality with conventional search algorithms. (**Top Right**) We propose a generative model MSA-Generator to produce informative MSA for these queries, offer a potential solution to such challenge.

Inspired by the generative prowess of language models (Raffel et al., 2020; Touvron et al., 2023; Chung et al., 2022; Chen et al., 2021), we see their potential beyond textual data. Through this lens, we liken protein sequences to text, aiming to innovatively generate virtual but constructive MSAs. These newly created alignments endow predictions with supplemental evolutionary information, amplifying the potential of protein structure predictions. In the domain of PSP, tertiary structure prediction emerges as a crutial challenge. It holds a central position in molecular biology, shedding light on intricate protein functions and interactions. And while secondary structure predictions (Rost & Sander, 1993) bring forth invaluable nuances, it's the tertiary predictions that offer a comprehensive view of a protein's intricate conformation.

For bio-tasks such as protein structure prediction, gene sequence alignment, RNA secondary structure determination, microbial community analysis, and evolutionary tree construction, where enhancing downstream performance necessitates multiple sequences, we introduce the sequences-to-sequences (seqs2seqs) generation task. Unlike the conventional sequence-to-sequence (seq2seq) tasks—e.g., machine translation, which requires a strict one-to-one correspondence between a source sequence $x$ and a target sequence $y$—seqs2seqs is designed for flexibility. The task aims to generate multiple coherent sequences from a given sequences. Each generated sequence preserves patterns from the input, but there's no strict correspondence between the input and output. Instead, we prioritize maintaining interconnected patterns across them. This design endows the task with a self-supervised nature due to its intrinsic adaptability.

In the context of protein, such flexibility enables easy extraction of a portion of the MSA as the source, with the remainder acting as the target. Harnessing search algorithms, our framework adeptly extracts source and target data from comprehensive protein databases, paving the way for self-supervised pre-training. To our knowledge, this marks an initial step in tapping into self-supervised generative pretraining to bolster protein structure prediction accuracy.

We introduce MSA-Generator, a protein language model pre-trained using the seqs2seqs as its pretext task. Specialized in simultaneously generating multiple sequences, it effectively captures global structural information from input MSAs. This approach facilitates rapid, *de novo* sequence creation, improves inferior MSAs (see fig. 1), and boasts adaptability across various protein domains, adeptly navigating computational challenges.

To summarize the main contribution of this article:

- **Innovative unsupervised Seqs2Seqs Task Proposition:** We propose the unsupervised sequences-to-sequences (seqs2seqs) task, a promising approach for generating informative protein sequences, with potential applications extending to other areas like RNA. Integrated with search algorithms, this task streamlines generative pre-training by automating data retrieval from expansive protein databases.
- **Launch of MSA-Generator Model:** MSA-Generator, our state-of-the-art generative protein model, is uniquely devised to employ the seqs2seqs task on a self-curated dataset. It is optimized for multi-sequence generation, skillfully extracting global MSA insights, and has demonstrated adaptability across diverse protein domains.
- **Robust Empirical Validation:** We validate our approach's potency, showcasing significant improvements on AlphaFold2 and RoseTTAFold using CASP14 and CASP15 benchmarks. This signifies a practical leap forward in tackling the intricate challenge of protein folding. It also showcases the promise of seqs2seqs pretraining in the field of bioinformatics.

## 2 RELATED WORK

**Protein Structure Prediction**  Proteins, while diverse, are built from a mere 20 unique amino acids. Their physical structure, which determines their function and attributes, is pivotal to grasping the essence of life. While the field has witnessed significant advancements like AF2 (Jumper et al., 2021) and RoseTTAFold (Baek et al., 2021), challenges remain. The success of AF2 can be ascribed to its adept utilization of MSAs, constructed by search algorithms such as DeepMSA (Zhang et al., 2019), JackHMMER (Johnson et al., 2010) and MMseqs2 (Steinegger & Söding, 2017) across vast databases including UniRef (Suzek et al., 2007) and BFD based on a protein query sequence. Conversely, single-sequence prediction methodologies (Chowdhury et al., 2021; Lin et al., 2022; Chowdhury et al., 2022; Wu et al., 2022b) often underperform in comparison. However, for protein queries

devoid of extensive family representations, obtaining quality MSAs is challenging. Consequently, the proficiency of MSA-driven techniques diminishes. In this context, our proposed method leverages generative techniques to combat the paucity of homologs in protein sequences, presenting a solution when traditional techniques falter.

**Protein Language Models**    Language models, initially designed for language processing, have found a expanding role in bioinformatics, primarily for protein sequence representation. This uplift in performance largely stems from the adaptation of the masked language modeling (MLM) strategy, a concept inspired by BERT (Devlin et al., 2019). ProtTrans series (including ProtBert, ProtTXL, ProtXLNet, and ProtT5 (Elnaggar et al., 2021)) and the ESM models like ESM-1b (Rives et al., 2021) and ESM-2 (Lin et al., 2022) epitomize the influence of Transformer architectures in this domain. Further solidifying this connection, research (Vig et al., 2020; Rao et al., 2020) has unveiled ties between protein representations and contact maps, highlighting evolutionary patterns vital to AF2's triumph. Such revelations have realigned research interests towards MSAs from single sequence. Here, the MSA Transformer (Rao et al., 2021) stands out, harnessing MLM specifically on MSAs.

**Protein Sequence Generation**    Beyond masked language modeling (MLM), a variety of generative techniques exist, each with its distinct objectives and methodologies. For instance, Potts models (Figliuzzi et al., 2018; Russ et al., 2020) are crafted specifically for individual MSA sets from which they're derived (Zhang et al., 2022). However, their shortcomings in adapting to different MSA sets (Sgarbossa et al., 2022) have spurred the development of generative language models. An exemplar, ESMPair (Chen et al., 2022), constructs MSAs of Interologs by classifying sequences based on their taxonomic lineage. In contrast, both ProGen (Madani et al., 2020) and ProGen2 (Nijkamp et al., 2022) focus on single-sequence generation, sidestepping the integral component of MSA. Another research direction involves VAE-based models (Riesselman et al., 2018; McGee et al., 2021; Sinai et al., 2017), originally developed for mutation evaluation. The challenge of efficiently sampling from distributions to create diverse and long sequences limits their application to downstream tasks. A study by Sgarbossa et al. (2022) utilized the MSA Transformer in a repetitive mask-and-reveal methodology, which unfortunately led to a compromise in sequence diversity. Of notable mention is EvoGen (Zhang et al., 2022), aiming parallelly at producing virtual MSAs for Protein Structure Prediction. However, EvoGen uniquely operates as a meta-generative model, requiring guidance from Alphafold2 to hone its MSA generation prowess.

Despite the lengthy context associated with MSA Generation, our work **connects self-supervised learning with MSA generation**. We highlight generative MSA pretraining, introducing an unsupervised sequences-to-sequences task specifically tailored for efficient MSA generation. To the best of our understanding, this marks a significant stride in the realm of protein sequence generation.

## 3    SEQUENCES-TO-SEQUENCES GENERATIVE PRETRAINING

We present the Sequences-to-Sequences generation task and the methodology for automatic dataset construction in section 3.1. Details of the proposed MSA-Generator are provided in section 3.2. In section 3.3, we delve into the ensemble approach of MSA-Generator for optimizing the Protein Structure Prediction (PSP) task.

### 3.1    SEQUENCES-TO-SEQUENCES GENERATION FOR PROTEIN SEQUENCES

Recognizing the pivotal role of Multiple Sequence Alignments (MSA) and the challenge posed by sequences with sparse homologous matches in real-world databases, we present the Sequences-to-Sequences (seqs2seqs) Task tailored for synthesizing virtual MSAs. Unlike traditional sequence-to-sequence framework often seen in machine translation task, which dictate a rigid one-to-one mapping between source sequence $x$ and target sequence $y$, seqs2seqs adopts a more flexible approach. Rather than necessitating a direct match, the framework

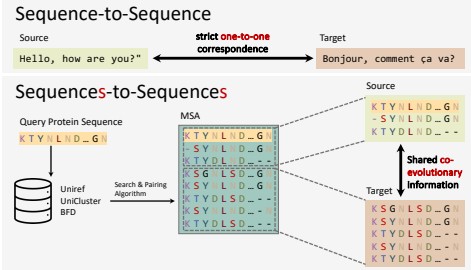

Figure 2: Difference between seq2seq and seqs2seqs and Automated Data Collection Process.

emphasizes that source sequences $X$ and target se-
quences $Y$ should exhibit **shared intrinsic patterns**. In the context of protein MSAs, this pattern alludes to co-evolutionary insights, necessitating the **capturing and amalgamation of extensive evolutionary data across and within the input sequences**, both horizontally and vertically.

The inherent adaptability of the seqs2seqs model permits the self-supervised nature of the task and seamless gathering of substantial quantities of source and target sequences from protein sequence databases. This is achieved by deploying sequence searching algorithms like JackHMMER (Johnson et al., 2010) and MMseqs2 (Steinegger & Söding, 2017). Our process began with selecting sequences from the UniRef90 database (Suzek et al., 2007) as initial queries. Subsequently, the JackHMMER algorithm (Johnson et al., 2010) was employed iteratively to identify homologous sequences within the database, based on the query sequences. This process was iterated until no additional sequences emerged, searching parameters are detailed in appendix C. For every batch of sequences retrieved, a random selection was made, designating query with some as the source $X$ and the remainder as the target $Y$, as illustrated in fig. 2. Notably, the assurance of co-evolutionary relationships is intrinsically facilitated by the search algorithm's mechanism.

## 3.2 SEQUENCES-TO-SEQUENCES ARCHITECTURE

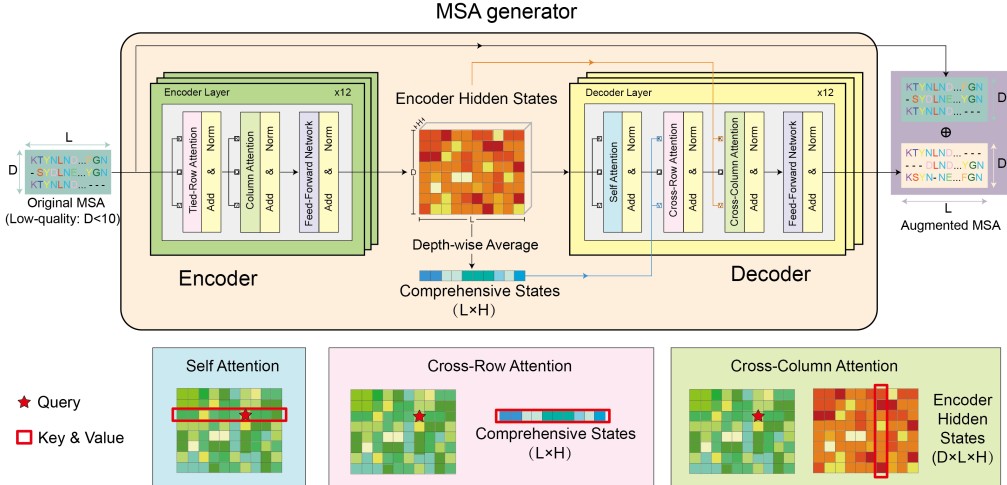

Figure 3: **MSA-Generator Overview** (Top) Overview of the architecture, processing pipeline, and module attention operations. (Bottom) Illustration of the attention mechanism. A red star represents a single query position, and the red boxes indicate keys and values utilized in attention processing and calculations.

We pretrain a transformer-based model (Vaswani et al., 2017), denoted as MSA-Generator, via the unsupervised Sequences-to-Sequences task. The MSA-Generator framework incorporates an encoder-decoder structure. The encoder contextualizes the input MSA data, while an auto-aggressive decoder produces sequences derived from this context (refer to section 3). To capture expansive evolutionary information from the input MSA both horizontally and vertically, the encoder integrates the tied-row and column attention mechanism (Rao et al., 2021). As the decoder concurrently generates multiple sequences—interacting with each other and the input MSA—it is enhanced with two additional modules beyond the conventional transformer. The *Cross-Row Attention* is designed to efficiently acquire a global representation by amalgamating comprehensive states. Meanwhile, to emphasize the vital conservative trait of amino acids (Dayhoff et al., 1978; Henikoff & Henikoff, 1992; Jones et al., 1992), we introduce the *Cross-Column Attention*, which, during the generation of the token at time step $t$, directs its attention to the $t$-th token of all input sequences.

**Tied-Row Attention** Building upon the foundation laid by the MSA Transformer (Rao et al., 2021), we incorporate a shared-attention mechanism. This is achieved by aggregating the attention map weights across each sequence from MSA $\in \mathbb{R}^{D \times L}$ prior to applying the softmax function. Notably, each sequence utilizes the same attention weight matrix. For the $d$-th row, the associated query, key, and value matrices are denoted as $Q_d$, $K_d$, and $V_d \in \mathbb{R}^{L \times h}$, respectively. These matrices are derived

via three distinct learned projection matrices. The computation of the shared attention weight matrix is formulated as:

$$W_{TR} = \text{softmax}\left(\sum_{d=1}^{D} \frac{Q_d K_d^T}{\lambda(D, h)}\right) \in \mathbb{R}^{L \times L} \tag{1}$$

In this context, $\lambda(D, h) = \sqrt{Dh}$ serves as the square-root normalization. This normalization is crucial in mitigating potential linear scaling of attention weights with the sequences. The resultant representation for the $d$-th row is obtained through $W_{TR}V_d$.

It's important to highlight that, in our decoder, we deliberately bypass the tied-row attention. This decision aids in maintaining diversity in the generated sequences. Instead, we lean towards a conventional self-attention mechanism.

**Cross-Row Attention**    Contrary to tasks like machine translation, where the target attends only to a single input during decoding, the essence of seqs2seqs lies in discerning intrisic patterns common to both source and target sequences. This necessitates a holistic comprehension of the input, implying that when generating a sequence, the decoder should attend to the entirety of the input. A naive concatenation would yield a representation with dimensions $\mathbb{R}^{D \cdot L \times h}$, rendering it computationally expensive and thus impractical.

To address this, we introduce an efficient strategy that calculates the depth-wise average pooling of encoder hidden states $H_{enc}$, represented as $H_c = \frac{1}{D} \sum_{d=1}^{D} H_{enc}^d \in \mathbb{R}^{L \times h}$. This serves as a global representation of the input and is crucial for cross-attention during decoding. Here, $K_c = H_c W_k$ and $V_c = H_c W_v$ signify the key and value matrices, while $Q = X_{dec} W_q$ stands for query matrix projected from decoder hidden states $X_{dec}$. The Cross-Row attention is:

$$\text{CR-Attention}(Q, K_c, V_c) = \text{softmax}(\frac{Q K_c^T}{\sqrt{h}})V_c \tag{2}$$

Each sequence generation process can access comprehensive information, attending to the same keys and values, mirroring the co-evolutionary patterns of the input MSA simultaneously thus **permitting fast parallel generation of multiple sequences** (middle at bottom in fig. 3).

**Self/Cross -Column Attention**    In MSA, each column represents residues or nucleotides at a specific position across sequences, revealing conserved regions essential for understanding biological functions, structural stability, or evolutionary significance (Dayhoff et al., 1978; Jones et al., 1992; Henikoff & Henikoff, 1992). Drawing inspiration from the vertical-direction attention proposed in Ho et al. (2019), we introduce a self-column attention in encoder for a comprehensive representation akin to Rao et al. (2021), and a cross-column attention in decoder to capture conservation characteristics.

To facilitate both attention mechanisms, the representation matrix $X \in \mathbb{R}^{D \times L \times h}$ needs to be transposed prior to the execution of self and cross attention:

$$\text{Column-Attention}(Q_{col}, K_{col}, V_{col}) = \left(\text{softmax}\left(\frac{Q_{col} K_{col}^T}{\sqrt{h}}\right) V_{col}\right)^T \tag{3}$$

For the self-column attention, projections from $X^T$ yield $Q_{col}, K_{col}, V_{col} \in \mathbb{R}^{L \times D \times h}$. In contrast, for the cross-column attention, $Q_{col}$ is determined from decoder hidden states as $X_{dec}^T W_q$, whereas $K_{col}$ and $V_{col}$ are projected from encoder hidden states as $H_{enc}^T$ (see fig. 3 bottom right).

**Pre-training objective**    We employ the seqs2seqs task to pretrain MSA-Generator. For a given source MSA $X \in \mathbb{R}^{D \times L}$, the loss is computed with respect to the target MSA $Y \in \mathbb{R}^{D' \times L}$ as follows:

$$\mathcal{L}_{seqs2seqs} = -\frac{1}{D' \times L} \sum_{d=0}^{D'} \sum_{l=0}^{L} \log P(y_l^d | y_{<l}^d, X) \tag{4}$$

It's crucial to note that each sequence $y \in Y$ is generated referencing the entire source matrix $X$, and this generation occurs in parallel owing to the thoughtful design of the architecture.

Pretrained MSA-Generator adopts 12 transformer encoders/decoders with 260M parameters, 768 embedding size, and 12 heads. It's pretrained with ADAM-W at a $5e^{-5}$ rate, 0.01 linear warm-up, and square root decay for 200k steps on 8 A100 GPUs, batch size of 64, using a dataset containing 2M MSAs constructed as described in section 3.1.

### 3.3 GENERATION AND ENSEMBLE STRATEGY

During inference, for every query sequence $x$, we initially use a search algorithm to assemble a MSA denoted as $X$. This is subsequently inputted into MSA-Generator to produce MSA $Y$. The concatenated MSA $X \oplus Y$ serves as input for the subsequent task. Nucleus sampling (Holtzman et al., 2019), set with *top-p=50* and *top-k=10*, is implemented to foster unique sequences and curtail redundancy.

For the purpose of optimizing the PSP task and yielding informative sequences, we adopt pLDDT as our selection criterion, leveraging our ability for swift MSA generation. pLDDT (Kryshtafovych et al., 2019; Jumper et al., 2021) measures the accuracy of predicted inter-residue distances for each residue in a protein, serves as a confidence indicator, with elevated scores hinting at potentially more accurate predictions. Utilizing pLDDT, we enhance each MSA through multiple runs, computing corresponding pLDDT scores for each. The MSA with the premier pLDDT score is subsequently selected as the optimal ensemble result and employed to determine the prediction accuracy relative to the ground truth.

## 4 EMPIRICAL VALIDATION

### 4.1 SETUP

The tertiary structure of a protein is pivotal, directly determining its functionality. In structural biology, the tertiary structure not only reveals the overall conformation of a protein but also inherently includes insights from secondary structures (Rao et al., 2021; Jones, 1999), such as the arrangement and orientation of $\alpha$-helices and $\beta$-sheets, and from contact predictions (Wang et al., 2017), denoting the spatial interactions between amino acid pairs. In essence, the tertiary structure offers a holistic view, allowing direct inference of localized structural features and interactions between amino acids. Leveraging tools like AlphaFold2 lets us directly obtain this comprehensive structural data, thereby bypassing intermediary steps like secondary structure and contact prediction. Given the centrality of tertiary structure prediction in protein functional studies, we have prioritized this task and adopted Local Distance Difference Tests [1] (LDDT) (Mariani et al., 2013) and pLDDT as our metric for evaluating prediction accuracy.

Specifically, we assess MSA-Generator by comparing preotein tertiary structures predicted from PSP algorithm, namely AlphaFold2 and RoseTTAFold, with various input MSAs. Demonstrating the usefulness of generated MSAs and the efficacy of MSA-Generator.

**Benchmark & Dataset**  We employ CASP14/15 as our test set, a prestigious dataset that encompasses proteins from a broad spectrum of biological families. The creation of a vast protein structure prediction dataset is prohibitively expensive, and given that AF2 has already trained on all previously available structures, this dataset emerges as the best evaluation benchmark. It's important to highlight that **sequences from CASP14/15 aren't part of our pretraining dataset** (refer to appendix A for detailed explanation), and our evaluations precede the AF2 version updated with CASP14/15 information.

Our primary interest lies in challenging protein sequences devoid of homologues, rendering traditional search algorithms ineffective. For every query in our test dataset, we use JackHMMER to search within UniRef90, which contains 70 million sequences, in order to gather related homologues. We define two scenarios: (1) *artificial challenging MSAs* , where we purposefully pick the top 15 homologues for each test set query as an *artificial gold standard*. From these, 5 homologues are further sampled as the *artificial baseline*, offering a synthetic challenge. (2) *real-world challenging MSAs* , which includes 20 sequences from test set, each with homologues less than 20, significantly challenge PSP algorithms. All assessments are executed in a zero-shot setting.

---

[1]OpenStructure is used for LDDT calculation `https://openstructure.org/`

## 4.2 ARE VIRTUAL MSAS AS GOOD AS REAL MSAS ?

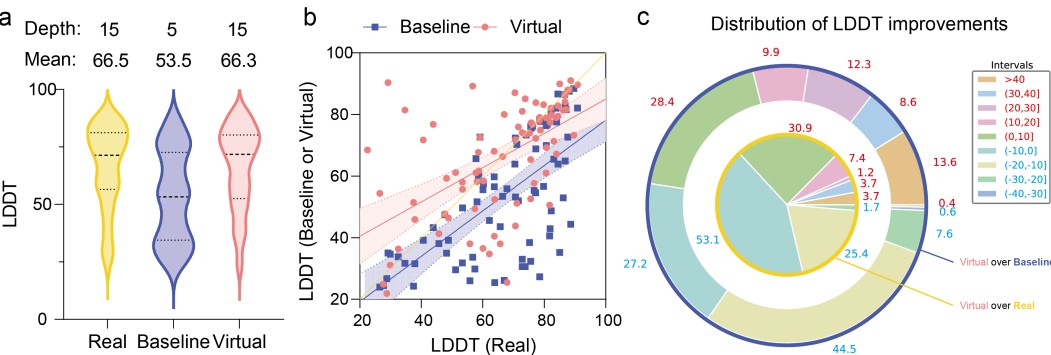

Figure 4: **Artificial Challenging Cases Results from AlphaFold2** (a) Violin plots of LDDT distribution. (b) x-axis represents LDDT of *artificial gold standard*, and the y-axis represents LDDT of *artificial baseline* and *artificial augmentation*. Dashed-line represents 95% confidence intervals (c) Pie chart of LDDT improvements in intervals. The inner circle represents a comparison with the baseline, while the outer circle represents a comparison with the real.

We employ *Artificial challenging MSAs* to thoroughly compare our generated virtual MSAs with conventional searched real MSAs to demonstrate that virtual MSAs can closely approximate real ones in downstream tasks. Specifically, for every *Baseline* MSA, we deploy MSA-Generator to produce a *Virtual* MSA that poses the same depth of the *Real* MSA. For each MSA, we employ an ensemble of three runs using the strategy outlined in section 3.3. Refer to fig. 4 for the results.

The fig. 4 (a) portrays that the LDDT distribution of the *baseline* suffers a sharp decline when reduced from 15 to 5 sequences. This underscores the importance of MSA quality for cutting-edge PSP algorithms. Yet, when supplemented by MSA-Generator's generative virtual MSA, the gap to the *Real* narrows considerably, reflecting a LDDT enhancement of 12.8. This underscores the effectiveness of our generated MSAs.

A more granular observation in fig. 4 (b) illustrates that the majority of *baseline* data points are below the diagonal, while most *Virtual* points sit above it, some even considerably outpacing the *Real*. fig. 4 (c) illustrates the statistics on improvements in intervals. It's evident that our generative virtual MSAs effectively improve results for 72.8% of protein sequences (over Baseline). Remarkably, nearly half of the generated virtual MSAs even outperform the real searched MSAs. This emphasizes the importance of generative MSAs and the potential of the seqs2seqs task in uncovering co-evolutionary patterns within bio-sequences. Among them, there are even 6 generative virtual MSA that surpass real MSA by more than 30 LDDT, including most notable T1032-D1 (**+46.02** LDDT), T1054-D1 (**+46.8** LDDT), and T1070-D2 (**+61.22** LDDT), suggesting that without generative virtual MSA, current PSA algorithm may fail on these queries. Comprehensive results for individual MSAs can be found in the table 5.

## 4.3 REAL-WORLD MSA CHALLENGES

Our ultimate goal is to produce high-quality MSAs for protein sequences with few homologues. Current search algorithms often fail to construct quality MSAs for these, making PSP algorithms similarly struggle with accurate predictions. The *Real-world challenging MSAs* evaluation is devised to test the efficacy of the seqs2seqs generative pretraining approach in addressing this challenge. For this, we curate sequences with fewer than 20 homologues using the search method detailed in section 4.1. For every identified MSA, we employ MSA-Generator to generate an MSA of identical depth across three independent runs. Subsequently, we measure the ensemble LDDT by inputting them to the PSP algorithm following section 3.3. For comparison, our benchmarks include the strong single-sequence folding technique, ESMFold (Lin et al., 2022) and OmegaFold (Wu et al., 2022a); we apply the same evaluation set up for the iterative unmasking strategy highlighted in (Sgarbossa et al., 2022); and generation with Potts models (Figliuzzi et al., 2018).

Table 1 presents the pLDDT and LDDT improvements achieved through various MSA generation techniques across different models. The single-sequence-based models lags behind MSA-based

| PSP Algorithm | CASP14 (avg. Det. = 6.1) | | CASP15 (avg. Det. = 7.4) | |
|---|---|---|---|---|
| | pLDDT | LDDT | pLDDT | LDDT |
| *single-sequence-based* | | | | |
| ESMFold | 43.3 | 41.9 | 46.0 | 53.4 |
| OmegaFold | - | 43.1 | - | 49.6 |
| *MSA-based* | | | | |
| RoseTTAFold | 63.5 | 51.3 | 62.6 | 52.1 |
| RoseTTAFold+Potts Generation | 63.2 | 48.9 | 94.4 | 51.0 |
| RoseTTAFold+Iterative Unmasking | 63.9 | 52.2 | 65.3 | 55.3 |
| RoseTTAFold+MSA-Generator | 68.9 $_{+5.4}$ | 56.3 $_{+5.0}$ | 69.0 $_{+6.4}$ | 58.4 $_{+6.3}$ |
| AlphaFold2 | 65.1 | 53.2 | 65.1 | 55.6 |
| AlphaFold2+Potts Generation | 63.8 | 50.9 | 64.5 | 52.6 |
| AlphaFold2+Iterative Unmasking | 65.2 | 54.6 | 69.5 | 57.3 |
| AlphaFold2+MSA-Generator | 71.6 $_{+6.4}$ | 57.5 $_{+4.3}$ | 73.7 $_{+8.6}$ | 63.7 $_{+8.1}$ |

Table 1: **Real-World MSA Challenges** average pLDDT and LDDT enhancement scores, averaged over 3 runs; avg. Det. represents average depth of MSA.

strategies. AF2 consistently outperforms RoseTTAFold in both metrics. Potts models, intriguingly, don't demonstrate a pronounced ability to produce effective MSAs. This is evidenced by their marginally reduced average performance in both metrics, echoing findings from (Sgarbossa et al., 2022; Rao et al., 2020). While iterative unmasking with MSA Transformer (Sgarbossa et al., 2022) can generate usable MSAs in certain scenarios, the MSAs it produces are often less diverse due to the inherent unmasking process, thus limiting its enhancement potential.

In contrast, our proposed method manifests marked improvements in both metrics across both models. This indicates that the MSAs generated by our technique are typically informative. Specifically, 75% of the MSAs were effectively enhanced, resulting in an average LDDT boost of 4.3 on CASP14 and 8.1 on CASP15. Remarkable LDDT enhancements were observed in T1093-D1 (with 3 homologous), moving from 45.5 to 70.77, and in T1113 (with 10 homologous), rising from 32.6 to 80.6. However, in specific MSAs, like T1094-D2 (7 homologous, pLDDT +2.8, LDDT -0.1), T1099-D1 (8 homologous, pLDDT +1.11, LDDT -0.5), and T1178 (15 homologous, pLDDT +2.2, LDDT -12.3), an elevation in pLDDT was offset by a decline in LDDT. This suggests that pLDDT might not always be a consistent indicator for selection strategy outlined in section 3.3. Detailed results for each individual MSA are available in the appendix D

### 4.4 RE-EVALUATING pLDDT AS A SELECTION METRIC

We sought to examine the effectiveness of pLDDT as a criterion. As previously highlighted, for specific proteins, improvements in pLDDT do not necessarily correlate with increases in LDDT. To delve deeper, we conducted an experiment where LDDT was directly calculated for each enhanced MSA in section 4.3. We then selected the highest LDDT as the output, bypassing the use of pLDDT as an intermediary metric. The detailed results are displayed in appendix D.

The disparity between pLDDT-based and LDDT-based predictions shown in fig. 5 suggests that pLDDT may not always be the best criterion. A noticeable gap exists between the two criteria

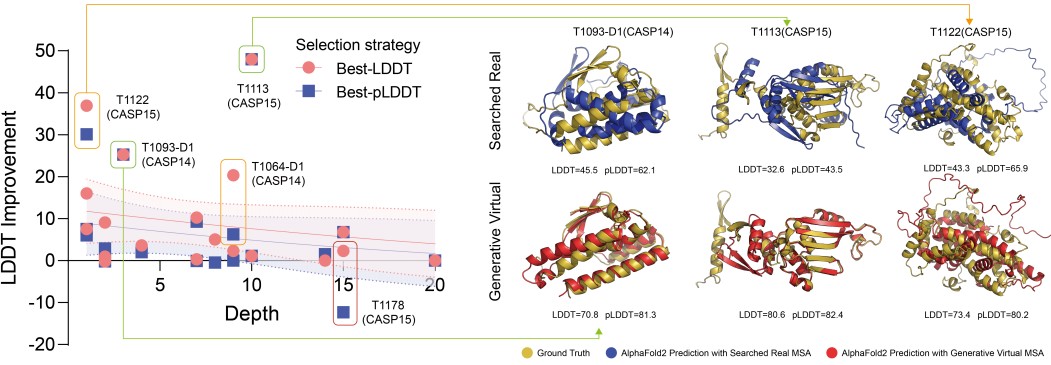

Figure 5: (Left) LDDT improvement selected by different criteria; (Right) Protein structure visualization with pLDDT and LDDT selected by Best-pLDDT.

(evident in the blue and red regions). For example, proteins T1064-D1 (depth=9) and T1122 (depth=1) exhibit significant gaps between scores chosen by LDDT versus pLDDT, highlighted in red boxes. Interestingly, for T1178 (depth=15), the highest pLDDT selection scores -12.3 against the baseline, while LDDT selection results in a +2.7. This implies that some generated MSAs, even with lower pLDDT scores, can enhance the LDDT. This indicates that our approach has untapped potential that could benefit from more nuanced selection criteria. The ideal situation would be for both predictions to produce the same significant improvement, as emphasized by the green boxes and visualized in fig. 5 (Right). Notably, MSA-Generator shows notable improvements for protein sequences with few homologs, emphasizing its utility in real-world protein folding challenges.

### 4.5 MSA DIVERSITY & CONSERVATION

We evaluated the generated MSAs themselves based on two key characteristics: **diversity** and **conservation**. All experiments setup are consist with section 4.3.To measure diversity, we analyzed the average Shannon Entropy across MSA columns. Compared to the Iterative Unmask method (Sgarbossa et al., 2022), our technique, as illustrated in fig. 6 (left), consistently yields higher Entropy, suggesting increased diversity. For conservation, we examined the Position-Specific Scoring Matrix (PSSM) (Altschul et al., 1997) of both original and MSA-Generator-generated MSAs. The PSSM gauges conservation of specific amino acids. The Pearson correlation coefficient between the searched real and generated virtual PSSMs, shown in fig. 6 (right), highlights the retention of amino acid conservation in generative virtual MSAs (see fig. 7 for visualization of conversation). The results underscore MSA-Generator's strength as an MSA generator, also showcasing the seqs2seqs framework's broader potential.

| #Runs | pLDDT | LDDT |
|-------|-------|------|
| 1 | 70.2 | 53.8 |
| 2 | 71.3 | 55.6 |
| 3 | 72.7 | 60.6 |
| 4 | 72.6 | 60.1 |
| 7 | 73.8 | 60.4 |
| 10 | 74.4 | 61.0 |

Table 2: Results of Ensemble Runs

| #AF | pLDDT | LDDT |
|-----|-------|------|
| 1 | 72.7 | 60.6 |
| 3 | 72.4 | 60.0 |
| 5 | 72.3 | 60.2 |

Table 3: Results of Augmentation Factor

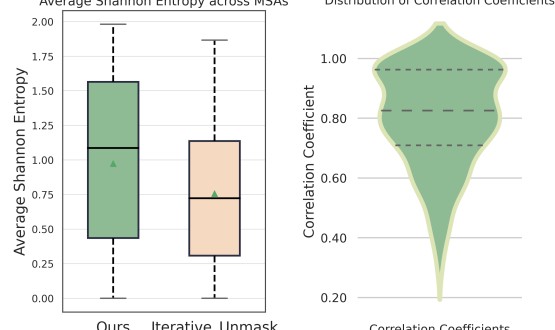

Figure 6: **(Left)** Box Plot of Averaged Shannon Entropy **(Right)** Violin Plot of Coefficients between Source PSSM and Generated PSSM

### 4.6 ABLATION STUDY

Following section 4.3 setup, we investigated the impact of Ensemble Runs and Sequence Augmentation Factor on PSP outcomes, as shown in table 2 and table 3. While additional ensemble runs lead to improved pLDDT scores, gains in LDDT plateau after three runs, even with higher computational expenses. Hence, we chose three runs to optimize performance and efficiency. Furthermore, a higher augmentation factor does not always yield better results; the identical input MSA might lack new insights, and extra sequences risk introducing noise.

## 5 CONCLUSION

We present an unsupervised seqs2seqs task, accompanied by an automated dataset construction pipeline, designed to pre-train MSA-Generator for simultaneous multi-sequence generation. Rigorous experimentation underscore the effectiveness, diversity, and conservation feature of generated virtual MSAs, amplifying the prowess of stalwarts like AlphaFold2 in scenarios where conventional methods come up short. Furthermore, our approach demonstrates generalization across a wide array of protein sequence families in a zero-shot fashion. Our findings highlight the immense promise of the unsupervised seqs2seqs task, pointing towards its prospective utility in a broader spectrum of bio-sequences, thereby amplifying its benefits.

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

## A    PRETRAINING DETAILS

We have compiled a pretraining dataset containing 2M MSAs from four databases: Uniref90 v.2022_04 using pipeline discussed in section 3.1. Specifically, for each MSA we randomly select 10-30 sequences and query as source $X$ and another 10-30 sequences as target $Y$ .

## B    MSA VISUALIZATION

Our goal is to explore the variations in MSA sequences using MSA-Generator. Accordingly, we depict the MSA's colored distribution in Fig 7 using Jalview. Observing the columnar distribution, it's evident that the produced MSA bears resemblances to the original sequences but introduces unique variations that encapsulate the external insights derived from MSA-Generator. This show the conservative feature reserved by the MSA and enhanced diversity as well.

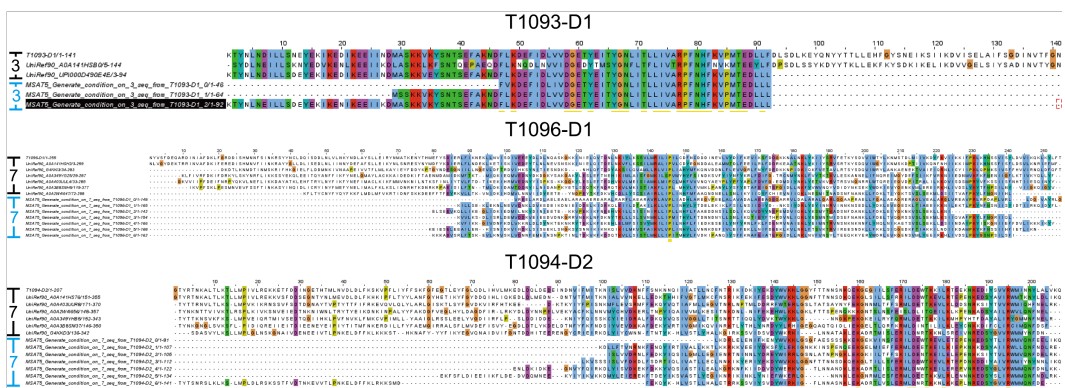

Figure 7: colored-distribution MSA, different colors represent different amino acids in protein sequence, from top to bottom is T1060s2-D1, T1093-D1, T1096-D1. since MSA-Generator augment one times more sequences, the top half of each diagram represents original MSA, and bottom half represent generated MSA

## C    SEARCH PARAMETER

We use JackHMMER to build pretraining MSA dataset. We adopt default parameters, including: -E = 10, -N=5, -Z=1000, –incE=0.01.

## D    DETAILED RESULTS

**Real-Word Difficult and Challenging Results**    Detailed results for Section 4.3 and 4.4 are presented in Table 4.

**Artificial Extreme Challenging Results**    Results for section 4.2 are shown in table 5 and table 6.

| ID | Depth | pLDDT | | | | LDDT | | | |
|---|---|---|---|---|---|---|---|---|---|
| | | Org | Run1 | Run2 | Run3 | Org | Run1 | Run2 | Run3 |
| **T1037-D1** | 4 | 36.04 | 40.68 | 36.04 | 36.04 | 24.09 | 26.08 | 24.09 | 24.09 |
| **T1042-D1** | 2 | 41.99 | 42.25 | 45.45 | 47.86 | 32.42 | 32.19 | 32.43 | 32.84 |
| **T1064-D1** | 9 | 60.34 | 66.93 | 63.1 | 65 | 31.25 | 37.48 | 31.34 | 51.36 |
| **T1074-D1** | 9 | 85.96 | 85.96 | 85.96 | 85.96 | 81.14 | 81.14 | 81.14 | 81.14 |
| **T1082-D1** | 10 | 92.31 | 92.79 | 92.31 | 92.31 | 87.37 | 88.49 | 87.37 | 87.37 |
| **T1093-D1** | 3 | 62.1 | 81.26 | 62.1 | 62.1 | 45.5 | 70.77 | 45.5 | 45.5 |
| **T1094-D2** | 7 | 90.54 | 90.54 | 93.34 | 91.47 | 77.12 | 77.12 | 77.02 | 76.47 |
| **T1096-D1** | 7 | 70.28 | 73.06 | 86.25 | 83.28 | 61.92 | 62.83 | 71.19 | 69.23 |
| **T1096-D2** | 2 | 45.55 | 50.01 | 54.55 | 50.24 | 34.07 | 36.56 | 36.98 | 34.66 |
| **T1099-D1** | 8 | 89.27 | 89.99 | 89.27 | 90.38 | 75.12 | 74.27 | 75.12 | 74.62 |
| **T1100-D2** | 2 | 42.16 | 47.27 | 42.16 | 42.16 | 34.93 | 35.87 | 34.93 | 34.93 |
| **T1113** | 10 | 43.5 | 82.4 | 76.4 | 77.0 | 32.6 | 80.6 | 78.9 | 78.0 |
| **T1119** | 2 | 93.4 | 94.5 | 92.4 | 93.6 | 91.1 | 90.8 | 90.4 | 90.0 |
| **1122** | 1 | 65.9 | 80.2 | 78.6 | 79.8 | 43.3 | 73.4 | 70.2 | 83.2 |
| **1125** | 15 | 49.6 | 54.4 | 54.8 | 53.2 | 38.2 | 45.0 | 45.0 | 43.2 |
| **1130** | 1 | 50.6 | 60.2 | 60.0 | 58.9 | 39.4 | 46.0 | 46.9 | 45.8 |
| **1131** | 1 | 40.6 | 46.0 | 45.4 | 46.2 | 32.6 | 48.6 | 38.2 | 38.6 |
| **1178** | 15 | 79.2 | 80.6 | 81.4 | 80.3 | 58.2 | 60.9 | 45.9 | 58.9 |
| **1194** | 14 | 93.4 | 94.1 | 93.4 | 93.4 | 90.2 | 91.7 | 90.5 | 91.7 |

Table 4: pLDDT (left) and LDDT (right) improvement over difficult MSA (depth≤20) of 3 runs.

| MSA-ID | Gold | Original | Aug1 | Aug2 | Aug3 | ENs | over original | over gold |
|---|---|---|---|---|---|---|---|---|
| T1024-D1 | 87.10 | 81.56 | 77.25 | 81.62 | 81.62 | 81.62 | 0.06 | -5.48 |
| T1024-D2 | 84.44 | 86.66 | 79.67 | 82.84 | 82.84 | 82.84 | -3.82 | -1.60 |
| T1025-D1 | 83.81 | 82.04 | 78.92 | 76.61 | 76.61 | 78.92 | -3.12 | -4.89 |
| T1026-D1 | 82.27 | 48.72 | 22.59 | 79.82 | 79.82 | 79.82 | 31.10 | -2.45 |
| T1027-D1 | 48.81 | 39.15 | 37.31 | 46.38 | 46.38 | 46.38 | 7.23 | -2.43 |
| T1028-D1 | 75.57 | 32.35 | 52.62 | 52.09 | 52.09 | 52.62 | 20.27 | -22.95 |
| T1029-D1 | 47.62 | 47.43 | 47.76 | 47.44 | 47.44 | 47.76 | 0.33 | 0.14 |
| T1030-D1 | 86.85 | 64.92 | 86.68 | 86.56 | 86.56 | 86.68 | 21.76 | -0.17 |
| T1030-D2 | 82.25 | 80.99 | 81.76 | 82.06 | 82.06 | 82.06 | 1.07 | -0.19 |
| T1031-D1 | 65.65 | 43.58 | 30.75 | 37.99 | 37.99 | 37.99 | -5.59 | -27.66 |
| T1032-D1 | 22.48 | 19.90 | 68.50 | 62.11 | 62.11 | 68.50 | 48.60 | 46.02 |
| T1034-D1 | 86.75 | 83.15 | 83.65 | 84.05 | 84.05 | 84.05 | 0.90 | -2.70 |
| T1035-D1 | 32.52 | 33.78 | 34.56 | 36.32 | 36.32 | 36.32 | 2.54 | 3.80 |
| T1036s1-D1 | 67.02 | 26.11 | 79.99 | 79.13 | 79.13 | 79.99 | 53.88 | 12.97 |
| T1038-D1 | 55.80 | 26.03 | 38.04 | 27.61 | 27.61 | 38.04 | 12.01 | -17.76 |
| T1038-D2 | 80.60 | 76.37 | 61.02 | 58.57 | 53.99 | 61.02 | -15.35 | -19.58 |
| T1041-D1 | 62.13 | 45.60 | 32.14 | 0.00 | 34.74 | 34.74 | -10.86 | -27.39 |
| T1045s1-D1 | 56.74 | 33.67 | 89.22 | 83.99 | 88.47 | 89.22 | 55.55 | 32.48 |
| T1045s2-D1 | 27.65 | 24.44 | 22.81 | 24.89 | 21.61 | 24.89 | 0.45 | -2.76 |
| T1046s1-D1 | 87.67 | 72.65 | 87.33 | 0.00 | 87.98 | 87.98 | 15.33 | 0.31 |
| T1046s2-D1 | 63.88 | 25.42 | 51.58 | 52.44 | 36.05 | 52.44 | 27.02 | -11.44 |
| T1047s1-D1 | 58.76 | 60.01 | 59.75 | 61.65 | 61.23 | 61.65 | 1.64 | 2.89 |
| T1047s2-D1 | 40.65 | 39.12 | 69.07 | 70.64 | 71.69 | 71.69 | 32.57 | 31.04 |
| T1047s2-D2 | 58.59 | 59.58 | 64.60 | 64.30 | 67.72 | 67.72 | 8.14 | 9.13 |
| T1047s2-D3 | 58.12 | 51.61 | 57.63 | 58.18 | 57.53 | 58.18 | 6.57 | 0.06 |
| T1048-D1 | 70.60 | 72.34 | 71.81 | 71.75 | 71.82 | 71.82 | -0.52 | 1.22 |
| T1049-D1 | 72.85 | 33.01 | 80.28 | 81.03 | 72.12 | 81.03 | 48.02 | 8.18 |
| T1050-D1 | 89.66 | 88.46 | 73.27 | 72.51 | 72.97 | 73.27 | -15.19 | -16.39 |
| T1050-D2 | 81.05 | 76.84 | 76.81 | 78.93 | 0.00 | 78.93 | 2.09 | -2.12 |
| T1050-D3 | 71.11 | 70.25 | 48.28 | 51.18 | 49.74 | 51.18 | -19.07 | -19.93 |
| T1052-D1 | 77.26 | 68.84 | 82.18 | 83.82 | 83.08 | 83.82 | 14.98 | 6.56 |
| T1052-D2 | 48.22 | 34.95 | 57.37 | 59.27 | 30.91 | 59.27 | 24.32 | 11.05 |
| T1052-D3 | 90.81 | 82.12 | 89.62 | 85.22 | 0.00 | 89.62 | 7.50 | -1.19 |
| T1053-D1 | 72.86 | 30.58 | 73.75 | 74.09 | 65.32 | 74.09 | 43.51 | 1.23 |
| T1053-D2 | 75.75 | 75.61 | 77.31 | 76.25 | 0.00 | 77.31 | 1.70 | 1.56 |
| T1054-D1 | 34.64 | 31.68 | 70.68 | 81.44 | 0.00 | 81.44 | 49.76 | 46.80 |
| T1055-D1 | 67.90 | 55.88 | 19.03 | 25.46 | 21.62 | 25.46 | -30.42 | -42.44 |

Table 5: *artificial challenging MSAs* results (1/2).

| MSA-ID | Gold | Original | Aug1 | Aug2 | Aug3 | ENs | over original | over gold |
|---|---|---|---|---|---|---|---|---|
| T1056-D1 | 75.61 | 76.11 | 64.62 | 65.39 | 62.93 | 65.39 | -10.72 | -10.22 |
| T1057-D1 | 86.18 | 52.96 | 81.38 | 80.30 | 81.04 | 81.38 | 28.42 | -4.80 |
| T1058-D1 | 45.72 | 38.10 | 37.47 | 41.31 | 39.76 | 41.31 | 3.21 | -4.41 |
| T1058-D2 | 51.18 | 29.59 | 61.13 | 60.41 | 60.73 | 61.13 | 31.54 | 9.95 |
| T1060s2-D1 | 75.25 | 38.44 | 72.84 | 74.01 | 0.00 | 74.01 | 35.57 | -1.24 |
| T1060s3-D1 | 80.69 | 69.93 | 78.72 | 78.31 | 78.89 | 78.89 | 8.96 | -1.80 |
| T1061-D0 | 60.24 | 57.51 | 54.74 | 0.00 | 58.36 | 58.36 | 0.85 | 0.85 |
| T1061-D1 | 37.50 | 24.28 | 51.45 | 46.94 | 50.60 | 51.45 | 27.17 | 13.95 |
| T1061-D2 | 53.34 | 31.82 | 58.05 | 50.32 | 55.92 | 58.05 | 26.23 | 4.71 |
| T1061-D3 | 81.92 | 42.92 | 77.44 | 77.34 | 76.29 | 77.44 | 34.52 | -4.48 |
| T1062-D1 | 59.10 | 72.66 | 72.61 | 59.58 | 0.00 | 72.61 | -0.05 | 13.51 |
| T1065s1-D1 | 88.59 | 44.32 | 90.31 | 91.00 | 0.00 | 91.00 | 46.68 | 2.41 |
| T1065s2-D1 | 70.46 | 63.90 | 80.77 | 87.09 | 83.49 | 87.09 | 23.19 | 16.63 |
| T1067-D1 | 78.55 | 27.33 | 79.05 | 75.29 | 77.91 | 79.05 | 51.72 | 0.50 |
| T1068-D1 | 85.05 | 35.20 | 89.55 | 88.93 | 89.92 | 89.92 | 54.72 | 4.87 |
| T1070-D1 | 53.44 | 49.68 | 55.69 | 52.55 | 52.73 | 55.69 | 6.01 | 2.25 |
| T1070-D2 | 29.16 | 35.04 | 87.50 | 86.84 | 90.38 | 90.38 | 55.34 | 61.22 |
| T1070-D3 | 74.12 | 73.32 | 73.56 | 73.50 | 0.00 | 73.56 | 0.24 | -0.56 |
| T1070-D4 | 73.83 | 30.62 | 74.39 | 84.71 | 85.07 | 85.07 | 54.45 | 11.24 |
| T1073-D1 | 76.17 | 76.19 | 76.79 | 0.00 | 76.11 | 76.79 | 0.60 | 0.62 |
| T1076-D1 | 83.10 | 65.52 | 71.31 | 68.61 | 68.35 | 71.31 | 5.79 | -11.79 |
| T1078-D1 | 77.22 | 54.56 | 0.00 | 64.61 | 72.25 | 72.25 | 17.69 | -4.97 |
| T1079-D1 | 84.51 | 63.88 | 79.44 | 69.34 | 68.08 | 79.44 | 15.56 | -5.07 |
| T1080-D1 | 63.68 | 59.76 | 67.91 | 67.67 | 68.08 | 68.08 | 8.32 | 4.40 |
| T1083-D1 | 78.49 | 78.26 | 78.02 | 77.48 | 77.72 | 78.02 | -0.24 | -0.47 |
| T1084-D1 | 86.64 | 86.47 | 49.09 | 86.03 | 86.25 | 86.25 | -0.22 | -0.39 |
| T1087-D1 | 80.76 | 77.32 | 37.14 | 39.73 | 69.27 | 69.27 | -8.05 | -11.49 |
| T1088-D1 | 26.48 | 24.00 | 33.92 | 34.10 | 54.40 | 54.40 | 30.40 | 27.92 |
| T1089-D1 | 83.99 | 77.55 | 64.70 | 64.30 | 67.15 | 67.15 | -10.40 | -16.84 |
| T1090-D1 | 76.73 | 55.34 | 57.65 | 0.00 | 56.06 | 57.65 | 2.31 | -19.08 |
| T1091-D1 | 43.89 | 43.26 | 70.35 | 71.45 | 76.76 | 76.76 | 33.50 | 32.87 |
| T1091-D2 | 81.76 | 40.56 | 50.57 | 0.00 | 47.30 | 50.57 | 10.01 | -31.19 |
| T1091-D3 | 71.58 | 52.00 | 75.93 | 0.00 | 75.98 | 75.98 | 23.98 | 4.40 |
| T1091-D4 | 79.76 | 56.64 | 83.20 | 0.00 | 78.01 | 83.20 | 26.56 | 3.44 |
| T1092-D1 | 63.77 | 56.56 | 33.97 | 38.95 | 40.48 | 40.48 | -16.08 | -23.29 |
| T1092-D2 | 70.61 | 53.20 | 43.76 | 26.54 | 43.80 | 43.80 | -9.40 | -26.81 |
| T1093-D2 | 29.91 | 34.69 | 28.20 | 28.12 | 31.08 | 31.08 | -3.61 | 1.17 |
| T1094-D1 | 28.74 | 26.73 | 21.89 | 21.63 | 20.72 | 21.89 | -4.84 | -6.85 |
| T1095-D1 | 60.06 | 55.57 | 35.95 | 36.50 | 32.78 | 36.50 | -19.07 | -23.56 |
| T1098-D1 | 57.45 | 29.82 | 49.36 | 45.94 | 36.76 | 49.36 | 19.54 | -8.09 |
| T1098-D2 | 37.88 | 31.58 | 45.70 | 45.89 | 45.36 | 45.89 | 14.31 | 8.01 |
| T1100-D1 | 57.78 | 65.70 | 60.39 | 60.01 | 61.10 | 61.10 | -4.60 | 3.32 |
| T1101-D1 | 88.14 | 87.57 | 87.95 | 87.64 | 86.52 | 87.95 | 0.38 | -0.19 |
| T1101-D2 | 78.74 | 77.81 | 43.27 | 42.09 | 74.91 | 74.91 | -2.90 | -3.83 |
| Average | 66.47 | 53.45 | 61.77 | 57.14 | 56.43 | 66.32 | 12.87 | -0.11 |

Table 6: *artificial challenging MSAs* results (2/2).

