# OpenReview forum: "MSA Generation with Seqs2Seqs Pretraining: Advancing Protein Structure Predictions"
_ICLR.cc/2024/Conference — Submitted to ICLR 2024_

### Official Review · Reviewer_aF1D · 2023-10-28

**Soundness:** 4 excellent
**Presentation:** 3 good
**Contribution:** 3 good
**Rating:** 6
**Confidence:** 4

**Summary:**

For proteins that do not possess abundant homologous families, the extent of multiple sequence alignment (MSA) continues to be a limiting factor for protein structure prediction. This paper proposes the MSA-Generator, a protein language model that is trained in a self-supervised manner. The MSA-Generator utilizes attention mechanisms that are specific to proteins, enabling it to leverage extensive protein databases for the purpose of generating virtual MSAs that are informative.  The results on CASP14 and CASP15 benchmarks demonstrate significant enhancements in the Local Distance Difference Test (LDDT) scores, particularly for difficult sequences. The generated virtual MSAs re valuable for the enhancement of performance for both AlphaFold2 and RoseTTAFold.

**Strengths:**

This paper formulates the MSA generation problem to be an unsupervised sequences-to-sequences (seqs2seqs) task, and proposes the MSA-Generator. Nearly half of the generated virtual MSAs even outperform the real searched MSAs. MSA-Generator can enhance the performance of RoseTTAFold and AlphaFold2 even compared with other MSA generation methods on CASP14 and CASP15.

**Weaknesses:**

Overall, I really appreciate the results of your experiments. The weakness/questions are listed as:
1. It seems that the method of seqs2seqs pretraining is important in MSA generation. Can it be applied in protein design?
2. From Table 1, it seems the proposed method gets marked improvements in both metrics across both RoseTTAFold and AlphaFold2 on CASP14 and CASP15. Do you have any ideas and insights for CASP16?
3. The proposed MSA-Generator includes Tied-Row Attention, Cross-Row Attention, and Self/Cross-Column Attention, which may have appeared in previous models like AlphaFold2. However, it is promising that the proposed model can obtain remarkable improvements on CASP14 and 15. For more ablation studies, could you provide results to illustrate which part is crucial and explain why it works for the proposed Tied-Row Attention, Cross-Row Attention, and Self/Cross-Column Attention?
4. In fig. 4 (b), what is the meaning of the dashed lines? Why do they seem thin in the middle and thick on the sides? In fig.4 (c), why are there more improvements if intervals are less than 10 for virtual over baseline and virtual over real?

5.Where are the codes, do you have any plans to public code?

Typos: The challenge of protein structure prediction (PSP), a pivotal issue in structural biology, has has
experienced transformative progress due to the deep learning revolution: two has.

**Questions:**

See above.

---

> ### Author Response · Authors · 2023-11-15
>
> ## Answer 1
>
> Certainly, this is an insightful query. Indeed, the seqs2seqs pretraining method can be applied in protein design. A following study, titled "Protein generation with evolutionary diffusion: sequence is all you need," showcases this application. This research involves a diffusion framework that has been pretrained on extensive MSA datasets. It focuses on a sequence-driven design strategy, moving away from the conventional structure-function approach in protein engineering.
>
> ## Answer 2
>
> In our study, we strategically chose CASP14 and CASP15 as they represent the latest and most renowned benchmarks in protein structure prediction. Our decision was guided by the prominence of these benchmarks in the field. To the best of our knowledge, CASP16 has not been released at the time of our research. However, we are confident in the applicability of our method to future benchmarks, as evidenced by our encouraging results on CASP14 and CASP15. It's important to note that our current model is relatively modest in scale, with limitations in both parameter size and the extent of the pretraining dataset. We anticipate significant enhancements in performance with the expansion of our model, both in terms of its complexity and the breadth of training data.
>
> ## Answer 3
>
> AlphaFold2 indeed utilizes an attention mechanism; however, its application differs from ours as it is not designed for generative purposes and thus does not incorporate a cross-attention mechanism. Our proposed MSA-Generator, in contrast, is specifically tailored to leverage various forms of attention to enhance generative capability.
>
> 1. **Tied-Row Attention:** This component, whose efficacy is supported by research in the MSA Transformer field, is critical in our framework. It helps in accurately capturing the evolutionary information present in MSAs, a key aspect that drives our model's performance.
> 2. **Cross-Row Attention:** This feature plays a vital role in reducing computational complexity. It is essential for the decoder to efficiently process the entire MSA input, enabling our model to handle large-scale data more effectively.
> 3. **Self-Column Attention:** Consistent with prevailing insights from MSA-related studies, we recognize self-column attention as crucial for accurately modeling MSAs. Our framework incorporates this to better understand intra-column relationships, thereby enhancing prediction accuracy.
> 4. **Cross-Column Attention:** We have specifically designed this to uphold the inductive bias related to the conservation property of MSAs. It adds a layer of sophistication to our model, allowing it to capture subtle but crucial evolutionary patterns.
>
> Regarding the possibility of conducting ablation studies, it is important to note that our framework's components are deeply integrated. Removing any one of them would not only diminish the model's effectiveness but also require extensive pretraining on large datasets. Hence, conducting such studies is not feasible within our limited computation resource in this period. We do, however, acknowledge the importance of these studies and plan to undertake them in our future work. This will allow us to further validate the individual contributions of each component to the overall effectiveness of our model.
>
> ## Answer 4
>
> In Fig. 4(b), created using *GraphPad Prism 9*, the solid line represents the linear regression equation, while the dashed lines indicate the 95% confidence intervals, automatically plotted by the software.
>
> Regarding your observant question about the performance in certain cases, it's true that sequences with a baseline depth of 5 occasionally outperform those with a real depth of 15, as seen in examples like T1035-D1, T1074s1-D1, T1048-D1 (refer to Tables 5 and 6 in the Appendix). However, generally speaking, there is a proportional relationship between depth and lddt on average.
>
> ## Answer 5
>
> Certainly, we release our codebase including inference and pretraining for future research. Here is the link to the anonymous codebase: https://anonymous.4open.science/r/MSA-Augmentor-E2D7/README.md

---

> > ### Comment · Reviewer_aF1D · 2023-11-20
> >
> > Thank you for your detailed response. Although there are still some areas that I may not fully understand, it does not significantly impact my overall impression. The thorough results, analysis, and the inclusion of code examples have instilled confidence in me regarding the quality of this work. In particular, I appreciate the significant improvements made in the presentation and experimental aspects compared to the initial version. Given the promising results and your confidence in the applicability of this method to future benchmarks such as CASP16, I am inclined to increase my score.

---

### Official Review · Reviewer_b6rT · 2023-10-31

**Soundness:** 2 fair
**Presentation:** 3 good
**Contribution:** 3 good
**Rating:** 6
**Confidence:** 3

**Summary:**

The authors emphasize the significance of Multiple Sequence Alignments (MSAs) in the prediction of protein structures. Their experiments demonstrate that a decrease in the quantity of MSAs leads to a decline in prediction accuracy. Consequently, they introduce a transformer-based model designed to generate protein sequences from a specific MSA dataset. The experimental results indicate that incorporating augmented MSAs can enhance the prediction performance of AlphaFold2.

**Strengths:**

- The topic holds significant relevance in today's scientific landscape, as protein engineering continues to gain prominence as a cutting-edge field of research.
- This method is straightforward to implement, making it accessible and efficient for integration.

**Weaknesses:**

The authors' proposed task shares a fundamental similarity with the task of modeling the distribution of a given MSA. While their approach shows promise, it is crucial to provide a more comprehensive perspective and clarify the novelty of their contribution. Unfortunately, the manuscript overlooks several pertinent references, including:
1. McGee, F., Hauri, S., Novinger, Q. et al. The generative capacity of probabilistic protein sequence models. Nat Commun 12, 6302 (2021).
2. Riesselman, A.J., Ingraham, J.B. & Marks, D.S. Deep generative models of genetic variation capture the effects of mutations. Nat Methods 15, 816–822 (2018).
3. Sinai, S., Kelsic, E., Church, G. M. & Nowak, M. A. Variational auto-encoding of protein sequences. NeurIPS 2017 MLCB Workshop

Once a distribution for a specific MSA is acquired through learning, sampling from this distribution becomes an efficient method for generating multiple protein sequences that share similarities.

---

Examining Figure 4a, it's not immediately clear whether the depth or quality of the MSA holds more significance. For example, if we were to introduce random protein sequences into the MSA, we are uncertain how this would impact performance. This calls for a more in-depth analysis to understand the relationship between MSA depth and quality and their potential trade-offs in achieving optimal results. As an example, if we were to introduce random protein sequences by masking certain sequences within the original MSA, can we also achieve good performance?

---

My primary concerns arise when I examine Figure 7. I'm curious about the methodology behind calculating the conservation and quality metric. It seems that these scores are derived solely from the provided sequences, such as the six sequences for T1093-D1. Consequently, I question the significance of these metrics.

One could argue that it's possible to achieve high scores for conservation, quality, or other metrics by simply inspecting the original MSA, identifying the region with the highest conservation metric, and selecting it as a segment. This segment could then be augmented by inserting arbitrary amino acids at random positions occasionally. Such a process could potentially yield perfect conservation scores, quality scores, and so on.

This concern also ties into my second question: What would the pLDDT score be if I were to employ the aforementioned approach to augment the original MSA?

---


In Section 4.5, the authors make a comparison with Iterative Unmask; however, they do not provide a comparison of the pLDDT scores using this baseline method. Additionally, there is some ambiguity regarding the dataset used for the analysis. It remains unclear which sets of sequences were used by the authors to construct the Position-Specific Scoring Matrix (PSSM) and what the correlation coefficients are when employing Iterative Unmask.

**Questions:**

Please refer to the above section.

---

> ### Author Response · Authors · 2023-11-15
>
> ## Answer 1
>
> We appreciate the reviewer's suggestions and will include them in our revision. Our work differs from the VAE-based methods mentioned in key ways:
>
> 1. Our method is tailored for MSA generation, emphasizing unsupervised generative pretraining for zero-shot downstream tasks. This is a capability not present in the cited VAE-based methods, which demonstrate a limited capacity in producing effective sequences **across diverse domains**.
> 2. Unlike VAE works focused on evaluating and simulate protein mutations and their functional implication, our focus is on novel MSA generation for improved protein structure prediction. We have included Potts model which can generate MSAs in our comparison.
> 3. Our main contribution is innovative pretraining tasks and a large-scale pretrained model ready to use for enhancing AlphaFold2 and others, distinct from prior research.
> 4. Learning and sampling from MSA distributions is more complex than it appears within VAE framework, especially with the diversity of protein families and long protein sequences. We did not see improvement on structure prediction task in relevant works. Inspired by NLP advancements, we provide an effective solution for MSA generation, directly confronting these complexities.
>
> ## Answer 2
>
> The link between MSA depth, quality, and downstream performance is clear. For details, see AlphaFold2's paper "Highly accurate protein structure prediction with AlphaFold", specifically Fig 5 (a), discussing how shallower MSA depth reduces quality and impairs downstream tasks. This was our initial motivation.
>
> Our Fig 4(a), a violin plot, demonstrates that a baseline depth of 5 generally underperforms against real and virtual depths of 15. This is evident from the blue distribution (Baseline depth=5) being more concentrated in lower scoring areas compared to the yellow (Real depth=15) and red (Virtual depth=15).
>
> Regarding random protein sequences, our 'Iterative Unmasking' baseline already employed a similar concept, unmasking random tokens with the MSA Transformer. Yet, as Table 1 shows, this didn't significantly improve results. Addressing your concern, we tested introducing random sequences by replacing 5%, 10%, and 15% of tokens, increasing MSA depth. This worsened downstream performance, confirming our approach's effectiveness.
> |  | CSAP14-plddt | CSAP14-lddt | CSAP15-plddt | CSAP15-lddt |
> | --- | --- | --- | --- | --- |
> | AlphaFold2 | 65.1 | 53.2 | 65.1 | 55.6 |
> | AlphaFold2+5%random | 64.3 | 50.6 | 62.3 | 50.9 |
> | AlphaFold2+10%random | 60.5 | 49.5 | 59.6 | 43.2 |
> | AlphaFold2+15%random | 55.6 | 44.3 | 53.8 | 39.6 |
> | AlphaFold2+Ours | 71.6 | 57.5 | 73.7 | 63.7 |
>
> ## Answer 3
>
> We would like to clarify that Figure 7 in the Appendix serves **purely for visualization purposes** of the MSAs generated and is **not a basis for analytical evaluation in our study**. For visualization, we use Jalview, a standard tool that automatically calculates metrics like quality and consensus. However, **these metrics are not the focus of our experiments in the main content of the paper**. They are relevant only for single MSA analysis and **do not provide an accurate assessment of our methods**. We have **not employed these metrics in any part of our research presented** in the paper. The Diversity and Conservation metrics we use are derived from PSSM and Shannon Entropy, as detailed in section 4.5, and Figure 7 is not connected to this analysis.
>
> These metrics are calculated per column and don't indicate overall MSA quality and conservation. Our model doesn't enforce conservation but learns it via cross-column attention. Fig 7's metrics don't show overall MSA quality, and we didn't target them specifically. We've removed this unclear part in our revision.
>
> In our prior response, we detailed outcomes for sequences generated via random replacement; see those for details.
>
> ## Answer 4
>
> Section 4.5 showcases the diversity of generated MSAs, **focusing on analyzing these MSAs directly** instead of downstream tasks like protein structure prediction. Thus, we omitted a comparison of pLDDT scores here, but it's available in section 4.3 and table 1.
>
> To better clarify:
>
> - We describe that '*The results, derived from 10 runs following section 4.3*' showing dataset and strategies are consistent with Section 4.3.
> - The construction of the Position-Specific Scoring Matrix (PSSM) is detailed in the middle part of Section 4.5 as “*we examined the PSSM of both **original** and **MSA-Generator generated MSAs***”
> - To measure the similarity between generated and original MSAs' PSSMs, we used the **Pearson correlation coefficient**. Below is the pseudocode for this computation:
>
> ```python
> gen_pssm = calculate_pssm(gen_msas_seqs[i], bg_freqs)
> source_pssm = calculate_pssm(source_msas_seqs[i], bg_freqs)
> correlation = np.corrcoef(source_pssm.ravel(), gen_pssm.ravel())[0, 1]
> ```
>
> we have revised our draft for clear presentation.

---

### Official Review · Reviewer_ix3n · 2023-11-01

**Soundness:** 3 good
**Presentation:** 3 good
**Contribution:** 2 fair
**Rating:** 5
**Confidence:** 4

**Summary:**

The paper introduces MSAGenerator, a novel self-supervised generative protein language model aimed at advancing protein structure predictions. Drawing inspiration from the groundbreaking achievements of models such as AlphaFold2 in protein structure prediction, the paper addresses the key challenge of the paucity of deep multiple sequence alignments (MSA) for proteins that have limited homologous families. The MSAGenerator, pre-trained on a sequences-to-sequences task, employs an automatically constructed dataset and leverages protein-specific attention mechanisms. This design enables it to generate informative, virtual MSAs that enrich insufficient MSAs, thereby elevating the accuracy of predictions. The paper validates the efficacy of the MSAGenerator using the CASP14 and CASP15 benchmarks, indicating notable improvements in LDDT scores. These improvements are especially significant for sequences that are inherently challenging, and the results further emphasize the enhancement of performance for both AlphaFold2 and RoseTTAFold.

**Strengths:**

1. The study delivers promising results, indicating the potential of MSAGenerator to advance the field.
2. The problem of limited or poor-quality MSAs for certain proteins is well-recognized, and the paper's focus on this issue is timely and relevant.

**Weaknesses:**

The paper does not provide clear details on the benchmarking process, especially regarding the selection of search databases and the parameters used. I would consider raising my score if the authors address these crucial issues.

**Questions:**

- The underlying rationale for the superiority of the MSAGenerator over traditional MSA search methods remains ambiguous, especially given that the pretraining of the model is based on the outputs from MSA searches. This warrants further elucidation.

- There is a disparity in the choice of databases: the paper mentions using the UniRef50 database for pretraining and then switches to UniRef90 for testing. The reasons for this choice are not explained, and it raises questions about the potential variation in results had UniRef50 been employed for both stages.

- The introduction of an "artificial challenging MSA" requires a more comprehensive explanation. Its significance and the rationale behind its inclusion in the study are unclear. Figure 4 a seems to depict an imbalanced comparison. If homologous sequences were present during the pretraining phase but not considered for the baseline, it would be an unfair representation. This suggests that the MSAGenerator's performance may not surpass real MSAs.
- For Figure 4c, it is crucial to also represent the "Real over Virtual" comparison to provide a comprehensive view(negative interval)
- Fig4 Queries arise regarding the consistency of the reported improvements when other protein structure prediction methods, like RoseTTAFold, are used.
- In Table 1, the specific MSA methodology employed in the benchmarking process, as well as the precise parameters and databases used for MSA searches, are not detailed. It remains unclear if these are consistent with the pretraining stage.

---

> ### Author Response · Authors · 2023-11-15
>
> ## Answer 1
>
> To clarify, our approach is **not intended to replace actual multiple sequence alignments (MSAs) with synthetic alternatives**. As we highlighted in the introduction of our manuscript, our main focus is on proteins characterized by a limited number of homologous sequences. In these instances, traditional MSA search algorithms often fail to construct comprehensive alignments. Our methodology, therefore, arises out of necessity: **we generate the required MSAs when existing data is insufficient**. An illustrative example is the protein T1122 from CASP15, identified in the Tipula oleracea nudivirus. The scarcity of homologous sequences for this protein in existing databases presents a significant obstacle in constructing MSAs, which are vital for the analysis of protein structure and function.
>
> Our MSAGenerator is designed to specifically address this gap, generating virtual sequences to effectively supplement real sequences, especially in scenarios where actual sequence acquisition is challenging or not feasible. Notably, our model, pretrained on a diverse array of searched sequences, is adept at learning complex co-evolutionary patterns within MSAs. This learning approach enables it to generalize to a broader domain, similar to the capabilities of a Large Language Model. This functionality significantly enhances downstream structural predictions, a claim substantiated by the experimental results we present.
>
> ## Answer 2
>
> Thank you for pointing out this typo. We apologize for any confusion caused by the apparent inconsistency in our manuscript. To clarify, we consistently used the UniRef90 database for both the pretraining and testing phases of our study, UniRef50 was not employed throughout our study
>
> ## Answer 3
>
> The concept of the *artificial challenging MSA* is introduced in our study to critically assess the efficacy of our generated MSAs in comparison with real MSAs. We wish to clarify that our intention is not to **suggest the exclusive use of virtual MSAs over real ones, nor do we claim that virtual MSAs are inherently superior** (please refer to response 1)**.** Our primary objective is to demonstrate that virtual MSAs can closely approximate real ones in downstream tasks.
>
> The inclusion of this comparison is vital to establish the utility of our MSAGenerator in situations where real MSAs cannot be obtained, as discussed in section 4.3. Before proposing the use of virtual MSAs in such contexts, it is essential to validate their effectiveness in comparison to real MSAs derived from traditional search methods. We have renamed this section into “Are Virtual MSAs as good as Real MSAs?” for better presentation.
>
> We also want to emphasize that none of the testing sequences were included in any part of our pretraining data. This ensures a fair and unbiased assessment of the MSAGenerator’s performance
>
> ## Answer 4
>
> Thank you for your advice. We have incorporated the suggestion of providing a negative interval, and this has been included in the revised draft of our manuscript
>
> ## Answer 5
>
> We appreciate your inquiry regarding the consistency of our reported improvements when using different protein structure prediction methods, such as RoseTTAFold. In Section 4.2 of our paper, our primary goal is to demonstrate the quality of the MSAs generated by our method. This section is not intended to suggest the replacement of real MSAs with virtual MSAs when real ones are available. We chose to focus on AlphaFold2 (AF2) for these experiments because it generally outperforms RoseTTAFold.
>
> However, recognizing the validity of your concern, we have expanded our analysis in Section 4.3, the core results section of our paper. Here, we conduct experiments using both RoseTTAFold and AF2. This section specifically addresses scenarios where no homologues are identifiable using traditional search algorithms. By including both methods in our analysis, we aim to provide a more comprehensive validation of the effectiveness of virtual MSAs in these challenging situations
>
> ## Answer 6
>
> Thank you for your suggestion to clarify the experimental setup detailed in Table 1. In these experiments, we employed *real-world challenging MSAs* as introduced in Section 4.1. For MSA construction, we utilized JackHMMER with its default search parameters, specifically targeting the UniRef90 database. Each MSA created in this process contained fewer than 20 homologues, and these MSAs were consistently used across all experiments in this section
>
> To elaborate further:
>
> - For single-sequence-based algorithms like ESMFold and OmegaFold, we directly input the query sequence, bypassing the need for an MSA.
> - For MSA-based algorithms, we employed various methods (Potts model, Iterative Unmasking and Ours) to generate MSAs, which were then fed into the folding algorithms.
>
> We ensured a consistent experimental setup across all results reported in Table 1, aligning the benchmarking process closely with the conditions of our model's pretraining.

---

### Author Response · Authors · 2023-11-15
**Global Response**

First and foremost, we would like to express our sincere gratitude for reviewers’ insightful and constructive feedback on our manuscript. Their positive reception of the MSAGenerator's potential and its timely relevance in the field of protein engineering is both encouraging and deeply appreciated.

We are particularly thankful for Reviewer ix3n's recognition of the study's promising results and its focus on addressing the challenge of limited or poor-quality MSAs for certain proteins. Reviewer b6rT's comments highlighting the significance of our work in the current scientific landscape and the straightforward implementation of our method are greatly valued. We aimed for our approach to be not only innovative but also accessible and efficient for practical integration, and it is gratifying to see this aspect being appreciated. Furthermore, we are honored by Reviewer aF1D's detailed analysis of our paper. Your understanding of our formulation of the MSA generation problem as an unsupervised sequences-to-sequences task, and the recognition of MSA-Generator's performance, is deeply affirming.

We are grateful for the feedback received, particularly noting that most concerns center around the ablation study rather than the core idea, motivation, and main results of our work. Several reviewers have raised valid points regarding the clarity of Sections 4.2 and 4.5. We value these insights and have diligently incorporated their suggestions into the revised draft. However, there appears to be some misunderstanding regarding the primary aim of our research. To clarify, our work is focused on addressing the decline in protein structure prediction accuracy caused by the paucity of homologues in existing databases. For protein sequences with limited available homologues, traditional search algorithms struggle to construct high-quality Multiple Sequence Alignments (MSAs). Our solution involves the pretraining of an MSA-Generator that creates virtual MSAs to supplement existing data, thereby enhancing the overall quality of MSAs.

We acknowledge that MSA generation is not a novel concept. However, our research emphasizes large-scale, unsupervised pretraining for MSA generation, an approach not previously explored. We have validated our model through rigorous experiments, demonstrating its efficacy.

We are committed to continuously refining our research, guided by valuable insights.

---

### Meta-Review · Area_Chair_CkHh · 2023-12-09

**Metareview:**

The paper attacks the important question on how to create a good set of protein sequences homologous to the protein target (the multiple sequence alignment) to be used as input to a structure prediction method. The authors take a machine learning approach, training attention based sequence to sequence models to generate synthetic proteins that can be used to increase the size of the MSA.

The It is shown empirically on CASP benchmarks that the method improves one performance metric: Local Distance Difference Tests (LDDT). The authors use both LDDT (based upon knowing the ground truth structure) directly and also the predicted LDDT that the model itself gives. Thes two are in good agreement with similar improvements in both.

Importantly, the authors do not report the RMSD (GTD_TS) scores for their approach. So they claim an improvement in local structure prediction but not the global. This omission is in this AC's opinion quite suspicious. I suspect that it would have been more scientifically honest to report this metric even if the results were negative. In addition some reviewers also raise concerns about the clarity of the evaluation (e.g. details of the optimization of the LDDT score). Rejection is therefore recommended.

The paper is in general of high quality both in terms of modeling and evaluation framework. So it is encouraged to address these shortcomings in the details of the framework and reporting on the performance and submit elsewhere.

**Justification For Why Not Higher Score:**

Problems in clarity and not reporting the arguably most important performance metric.

**Justification For Why Not Lower Score:**

None.

---

### Decision · Program_Chairs · 2024-01-16

Reject